# The Impact of Mortality Salience, Negative Emotions and Cultural Values on Suicidal Ideation in COVID-19: A Conditional Process Model

**DOI:** 10.3390/ijerph19159200

**Published:** 2022-07-27

**Authors:** Feng Huang, Sijia Li, Dongqi Li, Meizi Yang, Huimin Ding, Yazheng Di, Tingshao Zhu

**Affiliations:** 1Institute of Psychology, Chinese Academy of Sciences, Beijing 100101, China; huangf@psych.ac.cn (F.H.); lisj@psych.ac.cn (S.L.); lidongqi2000@163.com (D.L.); diyazheng97@163.com (Y.D.); 2Department of Psychology, University of Chinese Academy of Sciences, Beijing 100049, China; 3School of Child Development and Education, China Women’s University, Beijing 100101, China; ymz001017@163.com; 4School of Education, Renmin University of China, Beijing 100034, China; daicy9597@foxmail.com

**Keywords:** suicidal ideation, negative emotion, collectivism, individualism, terror management theory, big data analysis

## Abstract

As suicides incurred by the COVID-19 outbreak keep happening in many countries, researchers have raised concerns that the ongoing pandemic may lead to “a wave of suicides” in society. Suicidal ideation (SI) is a critical factor in conducting suicide intervention and also an important indicator for measuring people’s mental health. Therefore, it is vital to identify the influencing factors of suicidal ideation and its psychological mechanism during the outbreak. Based on the terror management theory, in the present study we conducted a social media big data analysis to explore the joint effects of mortality salience (MS), negative emotions (NE), and cultural values on suicidal ideation in 337 regions on the Chinese mainland. The findings showed that (1) mortality salience was a positive predictor of suicidal ideation, with negative emotions acting as a mediator; (2) individualism was a positive moderator in the first half-path of the mediation model; (3) collectivism was a negative moderator in the first half-path of the mediation model. Our findings not only expand the application of the terror management theory in suicide intervention but provide some insights into post-pandemic mental healthcare. Timely efforts are needed to provide psychological interventions and counseling on outbreak-caused negative emotions in society. Compared with people living in collectivism-prevailing regions, those living in individualism-prevailing regions may be more vulnerable to mortality salience and negative emotions and need more social attention.

## 1. Introduction

Since the coronavirus disease 2019 (COVID-19) outbreak, major adverse life events, including suicides due to a fear of contagion, have been reported in many countries and regions [1,2,3]. Some researchers worry that the ongoing pandemic may result in “a wave of suicides” in society [4,5]. Unfortunately, these worries are being continually validated as the outbreak continues. A meta-analysis involving 308,596 participants showed that suicidal ideation (SI) increased worldwide during the outbreak [6]. In China, the estimated lifetime prevalence of suicidal ideation in the general population is 3.9% [7], but it reached an alarming 16.4% during COVID-19 outbreak [8]. Another longitudinal study reported that Chinese adolescents’ suicidal ideation, suicide plans, and suicide attempts respectively increased by 7.2%, 3.9%, and 3.4% during the epidemic.

Moreover, the pandemic has triggered many negative emotions (NE) among the general population, such as terror [9], anxiety, depression [10], and panic attacks [11]. The psychological sequelae of the outbreak will probably persist for months and years to come [12]. Therefore, paying attention to the population’s mental health and providing targeted interventions and counseling during and after the outbreak is imperative. Suicidal ideation is a critical factor in identifying suicidal risks and realizing suicide intervention, and it is also an important indicator for measuring people’s mental health [13]. Therefore, to better provide mental healthcare and interventions for people after the COVID-19 outbreak, it is important to identify the contributory factors of suicidal ideation during and after the pandemic and its psychological mechanism.

Currently, the research into the causes of suicidal ideation mainly focuses on personality traits [14], adverse life events [15], and attitudes towards suicide [16]. Although the impacts of emotions on suicidal ideation have not drawn wide attention from researchers, the current studies indicate that negative emotions are significantly associated with suicidal behaviors [17,18,19]. In addition, compared with personality traits, negative life events, and attitudes towards suicide, whose impacts on suicidal ideation are indirect and subject to boundary conditions, negative emotions such as fear, anxiety, and depression have direct effects on suicidal ideation [15,20,21]. Given that negative emotions keep spreading among the public during the pandemic [22,23], it is necessary to examine whether they contribute to an increase in people’s suicidal ideation.

Since people’s reactions to negative emotions tend to be consistent across cultures and populations [24,25] and the impacts of negative emotion on suicidal ideation are more direct [26], the pandemic-related negative emotions may directly induce an increase in suicidal ideation across regions. It is important to find out the causes and boundary conditions of negative emotions to avoid the potential psychological consequences caused by negative emotions. The terror management theory (TMT) [27], one of the most influential psychological theories for studying threats and defensive behaviors, is widely used to discuss the impacts of mortality salience (MS) on people’s cognition, emotions, and behaviors. According to the TMT, human beings, as an advanced and intelligent species, are aware of their vulnerabilities and the inevitability of death (mortality salience) [28,29]. When epidemic diseases trigger mortality salience, natural disasters, and other death reminders, negative emotions such as terror and anxiety will spread among people, but “cultural worldviews” can help mitigate such negative emotions and help people gain symbolic immortality [28,29,30].

An increasing number of studies have proven the TMT principles. Once someone’s mortality salience is provoked, negative emotions such as anxiety, fear, and aversion can grab them, although this series of negative emotions can be mitigated via reinforced cultural worldviews [31,32,33]. The COVID-19 outbreak as a death reminder [34,35] has caused severe threats to people’s mental health, which could arouse negative emotions [22,23] and eventually result in a rise in suicidal ideation across the region [5]. Based on the TMT study and previous research on the relationship between negative emotions and suicidal ideation, we hypothesize the following:

**H1.** 
*Regional mortality salience is a positive predictor of suicidal ideation, with negative emotions acting as a mediator.*


Cultural values could play a moderating role in the process in which mortality salience arouses negative emotions. Individualism–collectivism is the most active dimension in cultural psychology studies [36]. Compared with individualism, which advocates freedom, independence, and self-realization, collectivism emphasizes the prioritization of the group over the individual. It stresses such concepts as family, nation, and state [37], which echoes the TMT’s idea of “building cultural worldviews to gain symbolic immortality”. Although it is inconclusive whether individualism and collectivism, usually classified into “cultural values”, are equivalent to the TMT’s “cultural worldviews”, studies have shown that collectivism can mitigate people’s negative psychological reactions to disasters [38,39]. Studies have also found that high collectivism can strengthen people’s psychological closeness and increase their prosocial behavior intentions after they experience traumatic events [40]. During the COVID-19 outbreak, multiple cases of suicide due to fear of contagion have been reported around the world [1,2,41], which may be related to the triggered mortality salience and sustained negative emotion [22,23]. Based on the TMT study and cultural psychology opinions, collectivism may reduce the level of negative emotion aroused by mortality salience and help mitigate their impacts on suicidal ideation, whereas individualism may work the other way around. In light of this, we hypothesize the following:

**H2a.** 
*Individualism is a positive moderator in the first half-path of the H1 model.*


**H2b.** 
*Collectivism is a negative moderator in the first half-path of the H1 model.*


The traditional psychological studies used retrospective surveys to measure people’s awareness, emotions, cultural values, and suicidal ideation. However, the social isolation and traffic control measures imposed during the pandemic made it impossible to carry out paper surveys. It is also difficult to effectively conduct Internet surveys in regions and among populations severely affected by the outbreak, as they may cause an additional burden to people. Big data from social media can be used to address the above difficulties. Weibo launched by Sina Corporation (Beijing, China) is the most influential Chinese online social network (OSN). Researchers can extract big data about OSN user behaviors (such as posting, commenting, and replying) to reveal the rich information in the data [42]. Compared with traditional methods, using OSN behavioral data for psychological studies can avoid the necessity of contacting the subjects [43], making it an ideal method for conducting scientific research during a pandemic. Previous studies have verified the effectiveness of using big data on social media to identify users’ emotions, levels of awareness, cultural values, and suicidal ideations [44,45,46].

Based on the terror management theory, the present study uses a social media big data analysis to examine the combined impacts of mortality salience, negative emotions, and cultural values on suicidal ideation during the COVID-19 outbreak.

## 2. Materials and Methods

Previous studies have found that many factors have been associated with an increased risk of suicide during the outbreak, such as regional lockdowns [47,48] and activity restrictions [49,50,51,52,53]. To avoid confounding factors as much as possible, we set the observation period from 31 December 2019 to 8 April 2020, considered the first phase of the outbreak in China. On the one hand, the Chinese government’s “regular epidemic prevention and control” plan for the COVID-19 outbreak [54] had not yet been implemented nationwide during this period, avoiding the involvement of confounding factors to some extent. On the other hand, the previous studies usually used a picture or text initiation paradigm to activate MS [31,55,56], which implies that the activation of MS by death reminders is important. Therefore, we considered 67 days to be enough to initiate the population’s MS, as the media continued to report on the infection and mortality of the outbreak.

The present study first involved downloading the public posts for all Weibo users during the observation period from a data pool containing 1.16 million active users [57]. Then, we used the psychological lexicons to identify their MS, cultural values, NE, and SI data. Finally, a conditional process model, proposed by Hayes and Rockwood [58] to evaluate mediating and moderating roles, was used to test the relationships between MS, cultural values, NE, and SI. All statistical analyses were conducted in R Studio. Figure 1 shows the entire process for the present study, from the data collection to statistical analysis.

### 2.1. Data Collection and Preprocessing

Sina Weibo is the most popular microblogging service for sharing and discussing individual information, life activities, and celebrity news in China [61]. The official application programming interface (API) is the gateway to access and download public content from Sina Weibo and is used to collect messages for all of mainland China [62]. Following the data collection processes of previous studies [46,53,63], we first downloaded the public posts of 1.16 million active Weibo users across 337 regions at the prefecture level in mainland China via Weibo’s API. Then, the active users during the outbreak were screened via the following conditions: (I) registered before 31 December 2019; (II) non-institutional, commercial, or celebrity accounts; (III) at least ten original posts (rather than retweeted posts) during the observation period. Finally, 108,914 active Weibo users were selected from 319 prefecture-level and above regions in mainland China for the present study.

After the data screening process, we used the TextMind system, a Chinese language psychological analysis system, to extract psycholinguistic features from active Weibo users’ original posts and extracted independent and linguistically labeled words [53,59,63]. In addition, previous studies have shown that people’s suicidal ideations are related to local economic conditions, such as inequality [64], unemployment [65], and poverty [66]. However, these data are not of the same statistical caliber, since the prefecture statistical bureaus have not published such data. Given this, we include here the per capita GDP information, which is considered to be an indicator of the general health of the economy [67], as a proxy control variable in the present study.

### 2.2. Psychological Lexicons

**The Simplified Chinese Version of LIWC (SCLIWC).** The present study used the SCLIWC [45] to calculate users’ MS and NE scores in different regions during the pandemic. The SCLIWC includes 155 keywords related to death, such as ‘qù shì’ (pass away), ‘lái shēng’ (afterlife), and ‘guǐ hún’ (ghost); and 846 keywords related to NE, such as ‘jīng kǒng’ (terror), ‘jiāo lǜ’ (anxiety), and ‘gǎn shāng’ (sadness). Previous studies have repeatedly verified the effectiveness of using SCLIWC to identify people’s psychological features [45,63,68].

**The Individualism–Collectivism Lexicon (ICL).** We used the ICL [60] to compute the individualism and collectivism scores of users in different regions. This lexicon includes 165 keywords related to individualism, such as ‘jìng zhēng’ (competition) and ‘wǒ’ (I or me), and to collectivism, such as ‘fēn pèi’ (allocation) and ‘wǒ men’ (we or us). By examining keywords related to individualism and collectivism used by Weibo users, we could effectively identify their cultural values [46,69,70].

**Chinese Suicide Dictionary (CSD).** The present study used the CSD [44] to calculate the SI scores of users in different regions. The CSD includes 586 keywords related to SI, such as ‘qiān guà’ (worry), ‘lún huí’ (reincarnation), and ‘yǒng bié’ (farewell forever). Previous studies repeatedly verified its effectiveness in detecting SI [44,71,72].

The present study used the above dictionaries to examine the frequency of keywords related to a specific psychological feature to quantify the psychological features. Formula (1) [46,73] was used to calculate keyword frequency.
(1)$$V_{r}=\frac{\sum K_{r}^{V}}{\sum W_{r}}$$

In Formula (1), ***V_r_*** represents the keyword frequency of a specific psychological variable in a region; ***K*** in the numerator stands for a keyword; ***W*** in the denominator stands for a word; the suffix ***r*** represents a region.

### 2.3. Statistical Analysis

In the present study, we constructed a conditional process model based on Model 9 proposed by Hayes [58,74]. In this model, MS is the independent variable (X) and the dependent variable is the SI (Y); NE is used as a mediator (M) for MS influencing SI; individualism (W) and collectivism (Z) are used as moderators for MS influencing NE. All variables in this study are continuous, and a schematic diagram of the conditional process model for MS impacting SI is shown in Figure 2. 

We first used the base R package [75] to compute the descriptive statistics and the Pearson correlation coefficient matrix for each variable of interest for the 319 regions. Then, in order to compare the results of our study with other research studies performed during the COVID-19 outbreak, we divided the dataset equally by time into two samples (prior period: 33 days from 31 December 2019 to 2 February 2020; later period: 34 days from 3 February 2020 to 8 April 2020) and examined the changes in all variables from the prior to later period using the matched samples *t*-test. Finally, the mediation package [76] was used to construct and test the conditional process model [58] for MS, NE, collectivism, and individualism as predictors of SI.

## 3. Results

### 3.1. Demographics

Among 108,914 active Weibo users, 40.98% were males and 39.25% were from Eastern China, which is considered richer than other regions in China [63]. The ages of users who registered their birth date in their profile ranged from 18 to 56 years, with the intended partial concentration in the 28 to 37 age group (62.67%). The demographic profile is depicted in Table 1. 

### 3.2. Correlational Analysis

Figure 3 shows the profiles for MS, NE, individualism, collectivism, and SI among the location regions. According to the correlation analysis, MS shows significant and positive correlations with NE and SI (*p* < 0.01), but non-significant correlations with individualism and collectivism (*p* > 0.05); NE shows significant and positive correlations with individualism, collectivism, and SI (*p* < 0.01); individualism shows a significant and positive correlation with SI (*p* < 0.01) but a non-significant correlation with collectivism (*p* > 0.05); collectivism shows a significant and positive correlation with SI (*p* < 0.01). Table 2 shows the descriptive statistics and a correlation matrix of the different variables.

### 3.3. Matched Samples t-Test

Table 3 shows the changes in MS, NE, individualism, collectivism, and SI from the prior period to the later period during the COVID-19 outbreak.

The results showed that compared to the beginning of the epidemic, MS, NE, collectivism, and SI scores in the 319 regions were increased significantly after the period of the COVID-19 outbreak (MS: t (318) = 30.28, *p* < 0.001; NE: t (318) = 30.04, *p* < 0.001; collectivism: t (318) = 32.23, *p* < 0.001; and SI: t (318) = 30.65, *p* < 0.001.) These results are consistent with previous studies; that is, the outbreak and persistence of COVID-19 cause a significant increase in people’s MS [35,77,78], NE [63], collectivism [35,79], and suicidal ideation scores [6,48,52]. Furthermore, we found that the change in individualism was non-significant (*p* > 0.05), which validates Oyserman, Coon, and Kemmelmeier’s opinion [80] that individualism and collectivism are two separate dimensions, whereby an increase in one does not imply a decrease in the other.

### 3.4. Multiple Regressions

According to the testing procedure of the conditional process model, we constructed two regression equations. First, we used MS, individualism, collectivism, and the interaction term to predict NE (Equation (1)). The results showed that MS is a positive predictor of NE (*B_simple_* = 0.44, t = 9.01, *p* < 0.001); the interaction effect of MS × individualism was positively significant (*B_simple_* = 0.10, t = 4.62, *p* < 0.001); the interaction effect of MS × collectivism was negatively significant (*B_simple_* = −0.11, t = −3.71, *p* < 0.001). The equation passed the F-test (F = 3.94, *p* < 0.001) and the independent variable was able to explain 37 percent of the variance in the dependent variable. Then, we used MS, individualism, collectivism, and NE to predict SI (Equation (2)). The results showed that both MS and NE are positive predictors of SI (MS: *B_simple_* = 0.28, t = 5.49, *p* < 0.001; NE: *B_simple_* = 0.38, t = 7.31, *p* < 0.001). The model passed the F-test (F = 48.84, *p* < 0.001) and the independent variable was able to explain 32 percent of the variance in the dependent variable. Table 4 summarizes all regression equations.

### 3.5. Conditional Process Analysis

**Testing the moderating role of individualism and collectivism.** As shown in Equation (1), for every SD increase in individualism, MS’s prediction of NE increases by 0.10; for every SD increase in collectivism, MS’s prediction of NE decreases by 0.11. To present the interactions between MS and individualism–collectivism, we conducted a simple slope test by dividing the individualism and collectivism results into four groups by M ± 1 SD (Figure 4). 

According to the test results, the high individualism–low collectivism, low individualism–low collectivism, high individualism–high collectivism, and low individualism–high collectivism groups show the varying effects of MS as a positive predictor of NE in descending order, as presented in Table 5.

**Testing the mediating role of NE.** We used the quasi-Bayesian parameter estimation method to test the mediating role of NE between MS and SI for different combinations of individualism and collectivism. The result shows that the mediating role of NE is significant for all four groups. The high individualism–low collectivism, low individualism–low collectivism, high individualism–high collectivism, and low individualism–high collectivism groups show the varying indirect effects of MS on SI with NE as a mediator in descending order, as presented in Table 6.

**Diagram of the conditional process model.** We plotted a model of the conditional process by which MS influences SI based on the results of multiple regression equations and quasi-Bayesian tests (Figure 5). As shown in Figure 3, MS significantly predicted SI through the mediating effect of negative emotions, which was enhanced by high individualism and weakened by high collectivism.

## 4. Discussion

The COVID-19 outbreak has triggered a serious psychosocial crisis worldwide, with suicide being a particularly serious topic [4,5,81]. From the perspective of the terror management theory, the present study revealed the relationship between mortality salience and suicidal ideation during the COVID-19 outbreak and the corresponding conditional process. On the one hand, the activation of mortality salience during the COVID-19 outbreak could generate persistent negative emotions in the social dimension, and this process could result in an increased risk of suicide; on the other hand, compared with high collectivism, the high-individualism-inclined people could be more susceptible to the impacts from mortality salience and negative emotions. The findings expand the application of the TMT in suicide intervention and provide insights into population mental healthcare during and after the outbreak.

The present study found that mortality salience was a positive predictor of negative emotions, which is consistent with previous studies [31,32,33]. According to TMT, when people perceive that they are exposed to a risk of infection—e.g., faced with the “death reminder” [27]—they will experience a range of negative emotions. During the COVID-19 outbreak, mass media became the major source of information about the epidemic [82]. It is worrisome that while appeals to people to take precautions are necessary, an unending barrage of news, especially misreporting (known as the “infodemic”), has proven to cause serious psychological consequences [83,84,85]. A report involving 38 million media reports showed that 84% of the misinformation about COVID-19 was neither challenged nor fact-checked before reaching the public [86], which can cause unnecessary fear in the population [87]. Therefore, on the one hand, we appeal to the relevant health authorities and professional associations to provide accurate and reliable information in a timely manner, which could be beneficial in dispelling people’s fear and uncertainty about COVID-19 [84]. On the other hand, we also call on the media to be more rigorous in reviewing and reporting information about the outbreak to reduce rumors. As mentioned in previous studies, sensational reports might boost Nielsen ratings, increase sales numbers, and fuel infodemics, but they add limited benefits to public health and welfare [84,86].

In the present study, we found that negative emotions are the most significant predictors of suicidal ideation, consistent with previous studies [17,26,88]. In some suicide-related theories, such as the cognitive theory of suicidal ideation and the interpersonal theory of suicide, negative emotions such as hopelessness, unbearability, and loneliness are the main predictors of suicide [89,90]. In addition, negative emotions usually act as mediators between other risk factors and suicidal behaviors. For example, previous studies have found that bad parent–child relationships [91] and posttraumatic stress disorder (PTSD) [21] are associated with elevated suicidal ideation, while the persistence of negative emotions transmits this effect. We found a similar mechanism in the present study, where negative emotions mediated the relationship between mortality salience and suicidal ideation. This means that if negative emotions widespread in society during and after the COVID-19 outbreak can be eliminated in time, the effects of other risk factors such as mortality salience on suicidal ideation could be reduced or even disappear. Considering this, we suggest that government bodies keep a constant track of public feelings and help relieve people’s negative emotions to avoid more severe psychological consequences.

We also found that individualism and collectivism played opposite roles in moderating the effects of mortality salience on a negative mentality; individualism amplified the effects of mortality salience on negative emotions and suicidal ideation, whereas collectivism reduced the effects of mortality salience on a negative mentality. These findings validate our hypotheses H2a and H2b. We consider that the moderating role of collectivism between mortality salience and negative emotions is similar to the buffering role of cultural worldviews, which may stem from the similarity between the traits of collectivism and TMT cultural worldviews. For example, both of them emphasize the connection between the individual (oneself) and the group (world) and the prioritization of the family, nation, and state over individuals to achieve symbolic immortality [27], which highlights the “immortal soul despite the decayed body” [92]. In contrast, individualism emphasizes individuality, independence, and self-realization [93]; individualists may suffer from more negative emotions and stronger feelings of loneliness during the COVID-19 outbreak, which may result in more serious mental issues. 

Furthermore, these findings imply that the individualism-inclined populations may suffer from a stronger negative mentality caused by the COVID-19 outbreak than collectivism-inclined populations. In China, teenagers (such as students or other adolescents) tend to endorse individualism more than seniors [94], and a study involving 56,679 Chinese participants showed that the impact of the epidemic on suicidal ideation was much greater in the younger group than in the older group [8]. Given this, we suggest that institutions such as universities and primary and secondary schools maintain the psychological condition of students, such as by conducting group counseling regularly to improve the psychological health of susceptible people. Moreover, in combination with the previous study [46], we also call on governments and the media to use “collectivistic encouragement” rather than “individualistic encouragement” when urging people to take the necessary precautions; for example, “wearing a mask can help keep you and your family safe from infection” may be a more humanistic and effective communication strategy than “wearing a mask can help keep you safe from infection”.

## 5. Limitations and Perspectives

The present study has the following limitations or perspectives. First, we selected research subjects only from social media, which could involve a potential sampling bias, as there may be more users from urban areas than from rural areas on Weibo, and the number of younger users may be larger than that of elderly users; therefore, future studies could incorporate more diverse sampling methods to balance the demographic information of the participants. Second, the definition of emotions in the present study was generalized; thus, future studies could subdivide negative emotions and further examine their effects on suicide risk by combining them with other suicide-related theories; for example, in cognitive theory, hopelessness and unbearability are the main predictors of suicidal ideation [89,95], while in interpersonal theory loneliness is a risk factor for suicide [90]. Third, we only use the GDP per capita as a proxy variable to control the economic status of the regions, and more specific macroeconomic indicators, such as the Gini coefficient, unemployment rate, and poverty index, could be considered in future cross-cultural studies. Finally, the present analysis is limited to the correlation level without being able to determine the causal relationship. Future studies could attempt to conduct further experimental research to examine the roles of mortality salience, negative emotion, and cultural values in suicidal ideation. 

The previous TMT studies have focused on how mortality awareness exists in people’s conscious and subconscious minds and influences their emotional or social behavior. Although suicide is not the major emphasis of TMT, mortality salience has been shown to trigger a range of negative emotions [31,32,33] associated with suicidal ideation [17,18,19]. The present study found that the activation of mortality salience during the epidemic could generate persistent negative emotions in the social dimension, and this process could result in an increased risk of suicide, while diverse cultural values could act as a buffer or intensifier. However, given that the COVID-19 outbreak has lasted over two years worldwide and that our data come from just the first four months in the same country, we suggest that future work could use the cross-cultural and longitudinal research paradigm to examine the lasting impacts of mortality salience, negative emotions, and cultural values on suicidal ideation.

## 6. Conclusions

The present study involved a social media big data analysis to examine the effects of mortality salience, negative emotions, and cultural values on suicidal ideation in the Chinese mainland during the COVID-19 outbreak. We found that (1) mortality salience was a positive predictor of suicidal ideation, with negative emotion acting as a mediator; (2) individualism was a positive moderator in the process in which mortality salience indirectly affected suicidal ideation; (3) collectivism was a negative moderator in the process in which mortality salience indirectly affected suicidal ideation. Our findings provided some insights into post-pandemic mental healthcare. We suggest that media coverage reduces descriptions of death and that the relevant authorities keep a constant track of negative emotions among the public and provide the necessary help. In addition, as individualism-inclined populations are more vulnerable to the negative emotion incurred by the outbreak than collectivism-inclined populations, we should pay more attention to them and provide timely help.

## Figures and Tables

**Figure 1 ijerph-19-09200-f001:**
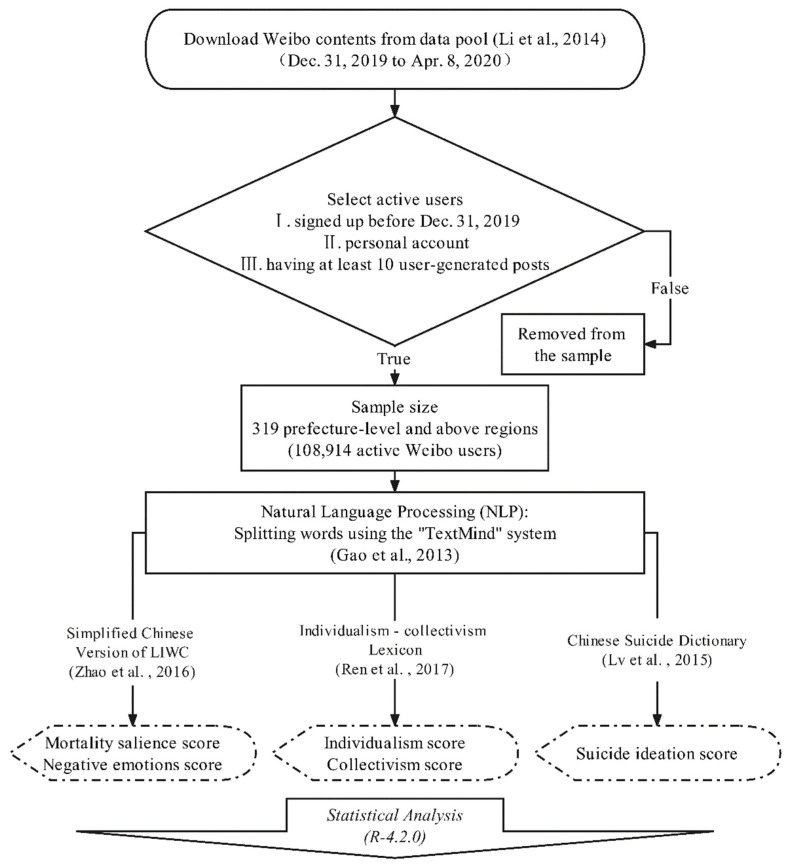
The research process for the present study [44,45,57,59,60].

**Figure 2 ijerph-19-09200-f002:**
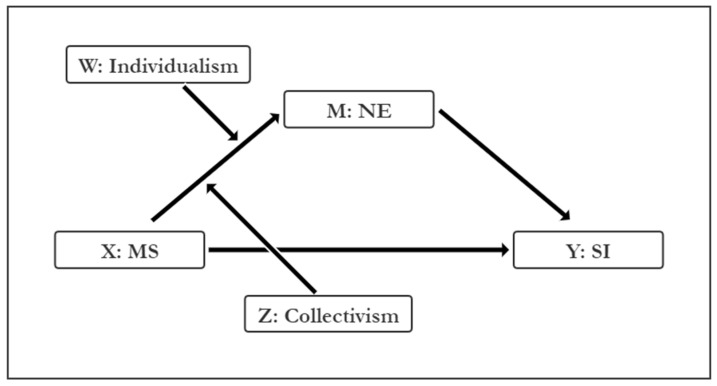
A conditional process model for MS impacting SI.

**Figure 3 ijerph-19-09200-f003:**
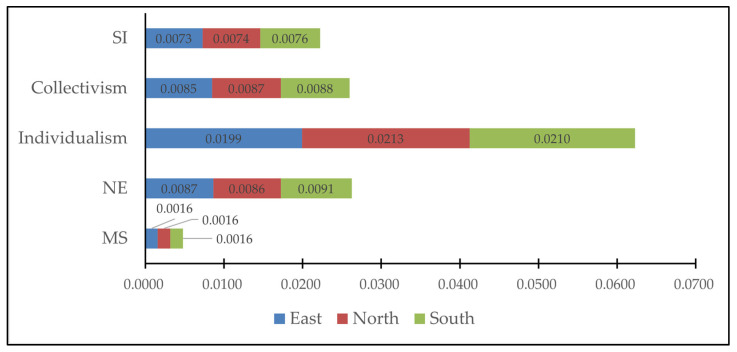
The profiles for MS, NE, individualism, collectivism, and SI among the different regions.

**Figure 4 ijerph-19-09200-f004:**
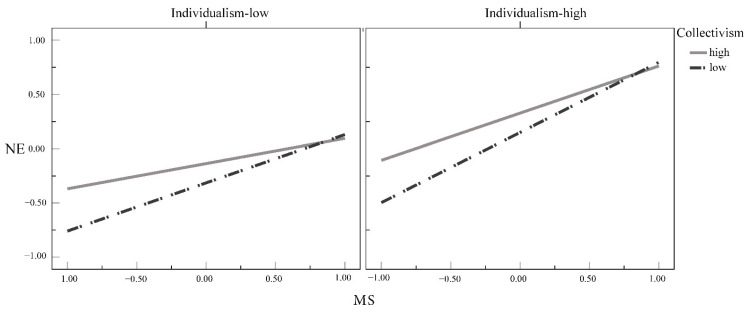
Simple slope test of the interactions between MS and individualism–collectivism. Note: Standardized values of variables are displayed.

**Figure 5 ijerph-19-09200-f005:**
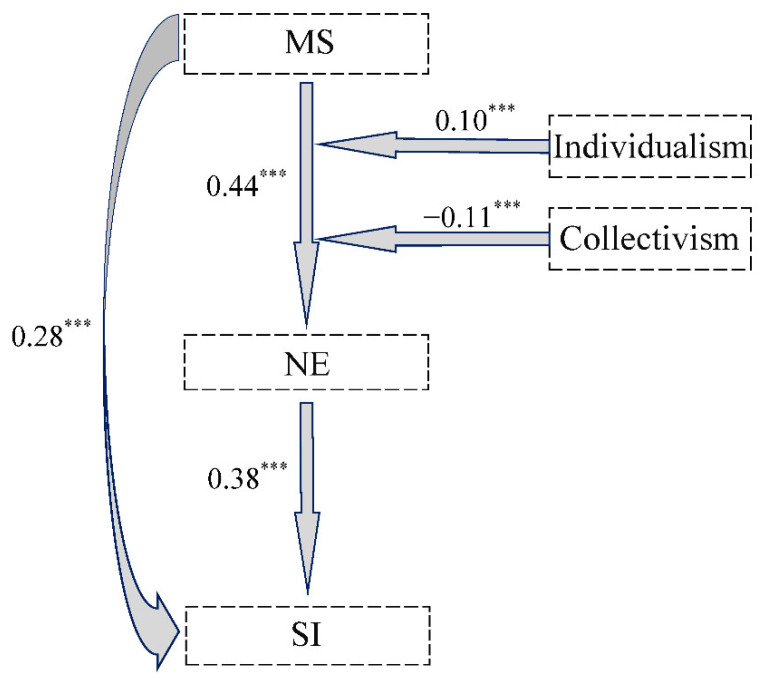
Conditional process diagram of how MS, NE, individualism, and collectivism impact SI. Note: Standardized coefficients are displayed; *** *p* < 0.001.

**Table 1 ijerph-19-09200-t001:** Demographic characteristics of selected participants.

Gender	male	44,629 (40.98%)
	female	64,285 (59.02%)
Age	18–27	29,381 (26.98%)
	28–37	68,256 (62.67%)
	38–47	9197 (8.44%)
	48–	2080 (1.91%)
Region of location	North China	27,905 (25.62%)
	East China	42,746 (39.25%)
	South China	38,263 (35.13%)
Total		108,914

**Table 2 ijerph-19-09200-t002:** Descriptive statistics and correlation matrix of different variables.

Variable	M	SD	1	2	3	4	5	6
1. GDP per capita (YUAN) ^1^	48,136.70	27,944.48	1					
2. MS	0.0016	0.0003	−0.10	1				
3. NE	0.0088	0.0012	−0.21 **	0.39 **	1			
4. Individualism	0.0209	0.0043	−0.11 *	0.01	0.36 **	1		
5. Collectivism	0.0087	0.0016	−0.11	0.08	0.19 **	0.05	1	
6. SI	0.0074	0.0014	−0.18 **	0.43 **	0.50 **	0.25 **	0.14 *	1

Note: ^1^ Since the statistics bureaus of some regions did not publish post-2016 GDP per capita information on their websites, the present study used 2015 GDP per capita information as a controlled variable; ***M*** is short for the mean, ***SD*** is short for the standard deviation; ***N*** = 319; * ***p*** < 0.05, ** ***p***
*<* 0.01.

**Table 3 ijerph-19-09200-t003:** The changes in all variables during the COVID-19 outbreak.

Variable	Prior Period		Later Period		t	df	Cohen’s d [95% CI]
	M	SD	M	SD			
MS	0.0013	0.0004	0.0019	0.0004	30.28 ***	318	[1.5851, 1.8054]
NE	0.0079	0.0014	0.0105	0.0015	30.04 ***	318	[1.5716, 1.7919]
Individualism	0.0209	0.0044	0.0209	0.0043	−0.30	318	[−0.1267, 0.0936]
Collectivism	0.0079	0.0017	0.0095	0.0017	32.23 ***	318	[1.6944, 1.9147]
SI	0.0065	0.0015	0.0088	0.0016	30.65 ***	318	[1.6057, 1.8261]

Note: The prior period represents the 33 days from 31 December 2019 to 2 February 2020; the later period represents 34 days from 3 February 2020 to 8 April 2020; M = mean; SD = standard deviation; df = degrees of freedom; *** *p* < 0.001.

**Table 4 ijerph-19-09200-t004:** Conditional process analysis of the effects of MS, NE, and cultural values on SI.

Independent Variables	Equation (1)			Equation (2)		
	(Dependent Variable: NE)			(Dependent Variable: SI)		
	β	SE	t	β	SE	t
GDP per capita	−0.12 **	0.05	−2.68	−0.07	0.05	−1.49
MS	0.44 **	0.05	9.01	0.28 ***	0.05	5.49
NE				0.38 ***	0.05	7.31
Individualism	0.23 ***	0.06	4.09			
MS × individualism	0.10 ***	0.02	4.62			
Collectivism	0.09	0.05	1.92			
MS × collectivism	−0.11 ***	0.03	−3.71			
Adj. *R*^2^	0.37			0.32		
F	3.94 ***			48.84 ***		

Note: Standardized regression coefficients are displayed; ** *p* < 0.01, *** *p* < 0.001.

**Table 5 ijerph-19-09200-t005:** Varying effects of MS as a predictor of NE among different combinations of individualism and collectivism.

Group	β	SE	t	95% Confidence Interval by Bootstrap	
				LLCI	ULCI
High individualism–low collectivism	0.65 ***	0.07	9.43	0.51	0.78
Low individualism–low collectivism	0.45 ***	0.05	8.53	0.34	0.55
High individualism–high collectivism	0.43 ***	0.07	6.66	0.31	0.56
Low individualism–high collectivism	0.23 ***	0.06	4.23	0.12	0.34

Note: Standardized coefficients are displayed; *** *p* < 0.001.

**Table 6 ijerph-19-09200-t006:** Indirect effects of MS on SI in different combinations of individualism–collectivism.

Groups	Indirect Effect Index	Boot SE	95% Confidence Interval	
			LLCI	ULCI
High individualism–low collectivism	0.24 ***	0.11	0.02	0.43
Low individualism–low collectivism	0.17 ***	0.07	0.02	0.28
High individualism–high collectivism	0.16 ***	0.07	0.03	0.29
Low individualism–high collectivism	0.09 ***	0.04	0.01	0.16

Note: Standardized coefficients are displayed; *** *p* < 0.001.

## Data Availability

To protect the participants’ privacy, the original posts used for the analysis are not publicly available but from the corresponding author at a reasonable request.

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
