# Peer review of "The Impact of Mortality Salience, Negative Emotions and Cultural Values on Suicidal Ideation in COVID-19: A Conditional Process Model"

_ijerph, 2022, doi:10.3390/ijerph19159200_

Round 1
Reviewer 1 Report
This is a novel approach to understanding components of the somewhat controversial terror management theory. It is a good use of social media to understand emotions and regional orientations to examine suicidal ideation and its association with mortality salience. The authors did a good job with making the manuscript readable with only moderate editing for the English language needed.
Comments:
1. It would be good to explain that conditional process analysis is a way of assessing both mediation and moderation, although this is not new. People have been doing mediated moderation and moderated mediation for decades. This is a new phrase for a common approach.
2. It is not clear how the Individualism-Collectivism Scale was used as two separate variables in the Equation 1 models. It seems as if this is the same scale and should be represented by a single continuous variable of the summed score, or maybe there is a cutoff score for classification purposes that has been validated. This require further explanation. If it is a single scale, then the variables created are not independent because if you are not in one category, you would need to be in the other. This needs explanation and without such an explanation, the remainder of the results are difficult to comprehend. This is also related to how the four groups were created.
3. Were the MS, NE, Individualism, Collectivism variables centered or standardized?
4. Usually the interaction is not included if the main effect is not significant. Although MSxCollectivism is significant, Collectivism is not significant at an alpha level of 0.05.
5. It is not necessarily a good idea to conduct a mediation analysis on cross-sectional data, but here it seems it is reasonable to assume that individualism and collectivism existed prior to the COVID-19 pandemic and seems reasonable, unless of course, the pandemic altered the degree of collectivism in each region, for better or worse.
6. Did you test whether a multilevel model might be more appropriate and use a random intercept or random slope for each of the 319 prefecture-level regions since you are trying to capture a cultural value? The individualism and collectivism could be represented as higher level variables.
7. It is not clear to me how terror management theory explains an increased risk of suicide ideation. It is my understanding that TMT explains group dynamic and ways in which people deny death or mitigate the fear. It seems on its surface, that suicide would not be an outcome of the theory. Can you better explain this connection for the reader? Is it possible that social isolation explains the SI better than TMT?
Reviewer 2 Report
The authors present an interesting study about the prediction of suicidal ideation based on
social media big data analysis.
There are several changes to be made should the article be published.
Results
Please present the results logically „3. Results 3.1. Descriptive Statistics”starts with correlational analysis.
Please present the results in Table 1 as required.
Please make corrections regarding writing as „The present study has the following limitations: First,” are writing correction –e.g. the verb can is not appropriate for what the authors what to suggest.
Also please check for inappropriate expressions such as „to the mentality of students” or „and NE are a positive predictor”, even panic attacks [8]. These are just illustrations.
Please discuss the results based on other theories such as the cognitive theory of suicidal ideation where hopelessness and unbearably are the main predictors of suicidal ideation, or please discuss it as a limitation.
Reviewer 3 Report
Thank you for giving me to review your manuscript. This manuscript is interesting and scientifically meaningful. Though I have a couple of observations & feedback as follows:
1. In the method section, the author should inform about Weibo – as it is a country-specific blogging app.
2. The authors also should describe the validity and reliability of ‘Weibo’s application program interface’ and ‘TextMind’ as these are not widely used and not globally validated.
3. There should be recall and response bias. The authors did not mention their measures to control these biases.
4. Some model fit information would be very helpful. The sample size is large, but the predictive power of the models is unclear.
5. A stronger call is needed re future research and policy implications in the discussion and conclusion section.
Reviewer 4 Report
This is a well-written article on the impact of mortality salience, negative emotions and cultural values on suicidal ideation in China during the first months of the Covid-19 pandemic. The findings are very useful for policy makers and could help form preventive strategies. However, I have some comments aimed at improving the manuscript.
Overall
- The manuscript requires some proofreading and editing
- There are many instances where the authors make certain statements but do not back them up with any academic references.
Introduction
- On line 55, the authors mention that “Given that NE keep spreading among the public during the pandemic”. This statement is not referenced to any research. I suggest that these types of affirmations are properly referenced.
- On lines 73-75 it is mentioned that “The COVID-19 pandemic as a death reminder… can provoke MS among people in a pandemic-battered region, which will arouse NE and eventually result in the rise of SI across the region.” Again, this is not properly referenced to any prior study.
- Another example of this is found on lines 89-91: “During the COVID-19 pandemic, multiple cases of suicide due to fear of contagion have been reported around the world, which may be related to the triggered MS and sustained NE.” I suggest this type of statement is cited with academic literature.
- The authors use big data from social media to identify users’ emotions, awareness, cultural values, and suicide ideation (SI). This has been previously verified. But in this particular study, I recommend the authors compare the prevalence they get for suicide ideation, with any previous and recent survey-obtained SI prevalence to see if this approximation through big data and social media is adequate or not, and how good or bad it is. If data from negative emotions and cultural values could be available with pre-Covid-19 data (from surveys or other sources), I also suggest that the authors compare the prevalence of these phenomena to measure the potential sample bias (that they correctly mention in the limitations) by using this methodological approximation.
- The authors should also mention or refer if suicide ideation or suicide attempt or even if suicide mortality has increased in China during the pandemic. This would better justify the research.
Methodology
- The observation period ran for four months. But given that the pandemic has lasted over two years, it has to be better justified why the authors used such a small period. This is relevant because a longer pandemic means a longer exposure to the researched risk factors and could have different consequences. Thus, I suggest the selection of this short period is better justified.
- In the introduction, I recommend the authors explain the GDP per capita and how it relates to the selected variables, to better justify its inclusion in the research. The following questions arise: Why use GPD per capita and not another macro-economic variable? Why not use inequality, unemployment, or poverty as a control variable? This issue must be properly addressed.
- In lines 142 to 144, the authors mention that “Given that individualism, collectivism and SI are all related to the regional economic level as proved by previous studies.” I suggest the authors refer (cite) to some of these previous studies.
- The authors estimate the Pearson correlation for every variable of interest for the 319 regions. The study would benefit from a brief description of these variables in the 319 regions. The reader does not know how the phenomenon behaved in those regions. Maybe a couple of maps could be useful to describe this behavior.
Results
- In this section, 3.2, the authors mention that two multiple regressions based on Model 9 of Hayes are estimated. These regressions are not presented or explained in the methodology. I suggest this is addressed. They have to explain with detail in the methods section, which are the dependent variables (how are they categorized or if they are continuous variables) as well as all the independent variables. Is not enough to mention that they are based on Hayes’s model.
Discussion
- In lines 262 to 264, the authors mention that “In the present study, though NE did not fully mediate the process in which MS effects SI, our findings are consistent with the conclusion of the cross-cultural [21], that is, NE, among all the risk factors, are the most significant predictor of SI.” The last part needs scientific references. It is a significant claim that NE are the most significant risk factors of SI that needs to be referenced with other literature.
Conclusions
- Given that the COVID-19 pandemic has lasted over two years, and the data used in this research comes from just the first four months, the authors should address how this longer exposure to the risk factors could affect SI. This should be at least addressed in the limitations of the paper.
Round 2
Reviewer 1 Report
Further English editing is required on lines 55, 73, 128, 234, 308 and other places in the discussion, however, it is much-improved since the first version. For example, on line 73, the TMT cannot really have opinions, but it can posit certain tenets based on the theory.
Figure 3 should have something about whether the numbers in the colored boxes are correlations, as they appear to be. They are quite small and it is working ;pondering whether statistical significance is relevant here with such small correlations, if indeed that is what they are.
Can or should the media truly reduce covering the death and infections of a pandemic? Maybe it is wise to be forthcoming about a disease that can kill you so that people will take the necessary precautions regardless of the effect on mental health as a public health measure.
Reviewer 4 Report
The authors addressed most of the recommendations made in the previous review. Yet, some limitations still remain.
The authors still do not explain why they chose the GDP per capita and how it relates to the selected variables. It is necessary to better justify its inclusion in the research. The following questions arise: Why use GPD per capita and not another macro-economic variable? This issue must be properly addressed.
The authors included some research in the begining of the paper to mention the behavior of SI in China. But these references are still not enough and leave unclear the o if suicide ideation or suicide attempt or even if suicide mortality has increased in China during the pandemic. This would better justify the research.
The observation period ran for four months. But given that the pandemic has lasted over two years, it has to be better justified and addressed in the limitations why the authors used such a small period, given that, as they state, negative emotions lasting and spreading for a long time, will severely harm people’'s mental health. Moreover, the authors state that 67 days is enough time to iniciate peoples MS, but this is not backed by any reference or prior study.
